# Making Scalable Meta Learning Practical

**Sang Keun Choe**[1][*] **Sanket Vaibhav Mehta**[1] **Hwijeen Ahn**[1] **Willie Neiswanger**[2]
**Pengtao Xie**[3,5] **Emma Strubell**[1,4] **Eric Xing**[1,5]
[1]Carnegie Mellon University  [2]Stanford University  [3]UCSD  [4]Allen Institute for AI  [5]MBZUAI

## Abstract

Despite its flexibility to learn diverse inductive biases in machine learning programs, meta learning (*i.e.*, learning to learn) has long been recognized to suffer from poor scalability due to its tremendous compute/memory costs, training instability, and a lack of efficient distributed training support. In this work, we focus on making scalable meta learning practical by introducing SAMA, which combines advances in both implicit differentiation algorithms and systems. Specifically, SAMA is designed to flexibly support a broad range of adaptive optimizers in the base level of meta learning programs, while reducing computational burden by avoiding explicit computation of second-order gradient information, and exploiting efficient distributed training techniques implemented for first-order gradients. Evaluated on multiple large-scale meta learning benchmarks, SAMA showcases up to $1.7/4.8\times$ increase in throughput and $2.0/3.8\times$ decrease in memory consumption respectively on single-/multi-GPU setups compared to other baseline meta learning algorithms. Furthermore, we show that SAMA-based data optimization leads to consistent improvements in text classification accuracy with BERT and RoBERTa large language models, and achieves state-of-the-art results in both small- and large-scale data pruning on image classification tasks, demonstrating the practical applicability of scalable meta learning across language and vision domains.

## 1 Introduction

Meta learning aims to learn the *inductive biases* (*e.g.* training data, neural architecture) of a machine learning program in such a way that a model trained with these inductive biases achieves optimal performance on user-specified *objectives* (*e.g.* fairness, quick generalization). This concept of meta learning can naturally be formulated as bilevel optimization, where the upper (meta) level problem encodes inductive biases and objectives, and the lower (base) level optimization problem represents the main machine learning program of interest, such as image classification or language modeling. Depending on the design of inductive biases and objectives, meta learning has found many applications in machine learning, including hyperparameter optimization [16], data optimization [21, 58], neural architecture search [38, 69], learned optimizers [43, 44], and few-shot learning [14, 51].

Following its versatility, numerous algorithms have been proposed to solve meta learning. Among them, gradient-based meta learning (GBML) has in particular gained considerable attention, due to its capability to optimize a wide range of *high-dimensional* inductive biases in an *efficient* manner. For example, MAML [14] finds optimal initialization weights (inductive bias) that achieve quick generalization to new tasks (objective), and L2RW [54] optimizes training sample weights (inductive bias) to achieve robustness against label noise (objective). However, the above benefits of GBML oftentimes get overshadowed by its poor scalability in practice, especially under the recent trend of large models [5, 33, 50], which arises due to several factors. First, many GBML algorithms [14, 40, 51] require inversion of a large Jacobian matrix, which suffers from both algorithmic instability as well

---

[*]Correspondence: `sangkeuc@andrew.cmu.edu`

| | Constant Memory | Jacobian Inverse Free | Adaptive Optimizer Support | (Efficient) Distributed Training Support | **Overall Scalability** |
|---|:---:|:---:|:---:|:---:|:---:|
| Iterative Differentiation [14, 15, 42] | ✗ | ✗ | ✔ | ✗ | ✗ |
| Recurrent Backpropagation [36] | ✔ | ✗ | ✔ | ✗ | ✗ |
| $T_1 - T_2$ [38, 41] | ✔ | ✔ | ✗ | ✗ | ✔ |
| Neumann Series [40] | ✔ | ✗ | ✔ | ✗ | ✔ |
| Conjugate Gradient [51] | ✔ | ✗ | ✔ | ✗ | ✔ |
| SAMA (ours) | ✔ | ✔ | ✔ | ✔ | ✔ |

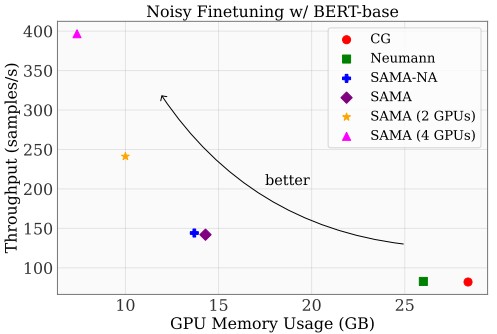
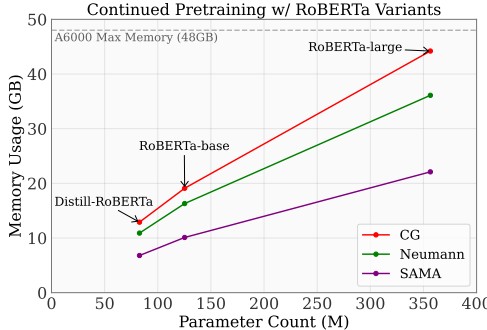

Figure 1: **Top:** Table showing a scalability comparison. **Bottom left:** Plot of throughput vs memory of different GBML algorithms on the *noisy finetuning of BERT-base* experiment. SAMA achieves better memory/compute efficiency overall given a fixed model, and the gap further widens by distributing compute across multiple GPUs with our efficient distributed training strategy. **Bottom right:** Plot of memory vs model size (*i.e.*, # of parameters) of different GBML algorithms on the *continued pretraining of RoBERTa* experiment. SAMA demonstrates the least significant increase in GPU memory usage with the increasing model size compared to baseline methods.

as exorbitant compute/memory costs. Second, most GBML research assumes that lower (base) level optimization is performed with SGD, whereas most large models, exemplified by Transformers [62], are by default optimized with adaptive optimizers like Adam [32]; consequently, the applicability of SGD-based GBML methods to large models trained with adaptive optimizers remains unclear. Finally, most GBML research to date has been limited to the single-GPU setup due to the lack of distributed training support [9, 20], which is essential in large-scale learning.

In this work, we endeavor to resolve the aforementioned scalability issues in GBML by co-developing algorithms and systems, and explore the initial potential of scalable meta learning in diverse applications. Our main contributions can be summarized as follows:

1. We investigate the root causes of substantial memory/compute costs, algorithmic instability, and a lack of distributed training support in GBML, all of which significantly limit its scalability, through a technical analysis of implicit differentiation. In doing so, we identify three major factors, namely (i) base Jacobian inversion, (ii) a lack of algorithmic adaptation for adaptive optimizers, and (iii) a need for the custom implementation of the backward pass of meta gradient computation, and discuss how each of them contributes negatively to the above limitations in depth.

2. Taking one step further, we propose an initial solution to each of the aforementioned issues by respectively (i) approximating the base Jacobian with an identity matrix, (ii) additionally expanding the meta Jacobian via the chain rule, and (iii) devising a novel communication strategy that efficiently uses the communication-computation overlap trick [35]. Combining all these solutions, we develop SAMA, a holistic and practically **S**cal**A**ble **M**eta learning **A**lgorithm.

3. We evaluate the scalability and the overall performance of SAMA on a multitude of large-scale meta learning benchmarks involving large language models (*e.g.*, BERT [31] and RoBERTa [39]) or large datasets (*e.g.*, ImageNet-1k [10]). Notably, SAMA showcases up to 1.7/4.8× increase in throughput and 2.0/3.8× decrease in memory consumption respectively on single-/multi-GPU setups compared to other baseline meta learning algorithms. In addition, we observe that SAMA-based data optimization consistently leads to improvements in text classification accuracy with large language models, and achieves state-of-the-art results in both small-/large-scale data pruning, demonstrating the initial potential of scalable meta learning.

## 2 Background: Gradient-Based Meta Learning

We begin by reviewing the basics of (gradient-based) meta learning in order to establish the key aspects that have limited its scalability. Mathematically, meta learning is commonly formulated as bilevel optimization as follows:

$$\lambda^* = \operatorname*{argmin}_{\lambda} L_{meta}(D_{meta}; \theta^*(\lambda))$$

$$s.t. \ \theta^*(\lambda) = \operatorname*{argmin}_{\theta} L_{base}(D_{base}; \theta, \lambda)$$

where $\lambda$ (respectively, $\theta$) are the parameters of meta (base) learners, $D_{meta}$ ($D_{base}$) are meta (base) datasets, and $L_{meta}$ ($L_{base}$) are meta (base) loss functions. An important implication of the above formulation is that meta learning changes the task of finding the optimal inductive biases from designing heuristics to designing meta optimization problems. As an example, consider the problem of finding the optimal inductive bias for fair classification given class-imbalanced training data. A traditional approach to this problem is to use a heuristic that reweights training samples inversely proportional to class frequencies. On the contrary, L2RW [54] designs a meta optimization problem by curating a small number of class-balanced data for the meta dataset $D_{meta}$ and setting the meta learner $\lambda$ to be importance weights for all training data. In short, unlike heuristic-based methods that explicitly specify "how to learn," meta learning methods only specify "what to learn" and let the meta learner automatically determine "how." From a programming paradigm perspective, such a difference can be understood as a transition from imperative to declarative programming.

While there are multiple approaches to solving meta learning, in this work we focus on *gradient-based* approaches due to their ability to efficiently solve high-dimensional meta optimization (*i.e.* $\dim(\lambda) \gg 1$) problems. Such an ability is essential given that the search space for inductive biases (*e.g.* importance weights for training data) can increase exponentially with recent large models and datasets. Concretely, GBML computes a meta gradient composed of two terms—the best-response Jacobian and direct gradient—with the chain rule, as follows:

$$\frac{\partial L_{meta}}{\partial \lambda} = \underbrace{\frac{\partial \theta^*}{\partial \lambda}}_{\text{best-response Jacobian}} \cdot \underbrace{\frac{\partial L_{meta}}{\partial \theta^*}}_{\text{direct gradient}} \tag{1}$$

Since the direct gradient (teal) computation is straightforward with the underlying automatic differentiation library, the major challenge in GBML lies in computing the best-response Jacobian (purple), of which two common solutions are iterative differentiation [14, 15, 16, 42] and implicit differentiation [25, 40, 49, 51]. Between these two, in this paper we adopt implicit differentiation as our baseline solution to GBML, as it achieves better computation and memory efficiency than iterative differentiation [19], both of which are vital in accomplishing our goal of scalable GBML.

The gist of implicit differentiation is that it calculates the best-response Jacobian by leveraging Cauchy's Implicit Function Theorem (IFT) and re-interpreting the base optimization problem from the perspective of fixed-point iteration given an iterative solver $u$, as follows:

$$\frac{\partial \theta^*}{\partial \lambda} = - \underbrace{\frac{\partial u}{\partial \lambda}}_{\text{meta Jacobian}} \cdot \underbrace{\left( \frac{\partial u}{\partial \theta^*} \right)^{-1}}_{\text{base Jacobian}} \quad \text{where} \begin{cases} \theta^* = \lim\limits_{t \to \infty} \theta_t \\ \theta_t = \theta_{t-1} - u(\theta_{t-1}; \lambda) \end{cases} \tag{2}$$

While exact implicit differentiation requires solving the base optimization problem to convergence $\theta^*$ by repeatedly applying an iterative solver $u$ (*e.g.*, SGD or Adam) to calculate base (blue) and meta (red) Jacobians, this is computationally impractical, especially in most large-scale learning settings. Therefore, researchers oftentimes approximate $\theta^*$ with a small number of unrolled update steps of $u$. This results in a solution that alternates gradient descent between base and meta optimization problems, where the base gradient is calculated with standard backpropagation and the meta gradient with Eqs. (1) & (2). Noting that many techniques have been developed to efficiently perform and scale up standard backpropagation, we deduce that the major challenges in scaling GBML lie in meta gradient computation, which will be discussed in depth in the next section.

## 3 Scaling Meta Learning

It has long been recognized that meta gradient computation in GBML suffers from a substantial compute/memory cost [40, 51], algorithmic instability [2, 12], and a lack of efficient distributed

training support [3, 6, 9], all of which can significantly limit its scalability. In this section, we first attempt to understand the above limitations at a technical level. Toward this end, we investigate three aspects in Eqs. (1) & (2), namely (i) base Jacobian inversion, (ii) algorithmic adaptation for adaptive optimizers, and (iii) a need for the custom implementation of meta gradient computation, and discuss how they lead to the aforementioned limitations. Next, we propose initial solutions for each of these issues, based on which we build a holistic and **S**cal**A**ble **M**eta Learning **A**lgorithm, SAMA.

## 3.1 Base Jacobian Inverse

**Problem**   Denoting the size of the base learner (*i.e.*, $\dim(\theta)$) as $n_b$, the computational complexity of the naive base Jacobian (blue) inversion in Eq. (2) is $\mathcal{O}(n_b^3)$. Since such cubic complexity is impractical even for small-sized base learners, practitioners typically utilize various linear systems techniques such as Neumann series [40] or conjugate gradient [51], and directly *approximate* $\left(\frac{\partial u}{\partial \theta^*}\right)^{-1} \frac{\partial L_{meta}}{\partial \theta^*}$. Given that the base Jacobian is a function of the Hessian matrix of the base optimization problem in GBML, these algorithms solve linear systems by iteratively performing Hessian-vector products. However, this Hessian-vector product computation contributes negatively to all three above limitations. First, while many automatic differentiation engines provide an efficient Hessian-vector product implementation like Pearlmutter's algorithm [48], its memory/compute cost is still prohibitive with larger models, as demonstrated in Fig. 1. Second, in most cases, we only have access to a stochastic estimation of the Hessian due to mini-batch sampling [34]. Hence, meta gradients obtained with noisy Hessian-vector products can be biased, which may result in training instability. Finally, the most efficient distributed training features, such as communication-computation overlap [35], are designed for first-order gradients, rather than the higher-order gradients involved in Hessian-vector products.

**Solution**   A simple solution for avoiding the aforementioned issues stemming from base Jacobian inversion is to approximate the base Jacobian with an identity matrix as follows:

$$\frac{\partial L_{meta}}{\partial \lambda} = -\frac{\partial u}{\partial \lambda} \cdot \left(\frac{\partial u}{\partial \theta^*}\right)^{-1} \cdot \frac{\partial L_{meta}}{\partial \theta^*} \approx -\frac{\partial u}{\partial \lambda} \cdot \frac{\partial L_{meta}}{\partial \theta^*} \tag{3}$$

Under the deep equilibrium model setting, Jacobian-free backpropagation [17] shows that such an approximation can be understood as preconditioning the original meta gradient. Our approximation also resembles approximating the Hessian as an identity matrix in one-step unrolling techniques (*i.e.* $T_1 - T_2$) from [38, 41]. While this approximation is exact when the iterative solver $u$ is naive SGD where $u = \frac{\partial L_{base}}{\partial \theta}$, we note that the base Jacobian does not necessarily equate with the Hessian when an adaptive optimizer is used in base optimization (more detailed discussion on this issue is deferred to Sec. 3.2). Furthermore, their methods calculate the meta Jacobian at initialization $\theta$ instead of at convergence $\theta^*$ due to their close connection to iterative differentiation [42], and thereby inhibit unroll steps larger than 1, unlike our approach. Considering that a larger number of unroll steps allows for less frequent computations of expensive meta gradients, our implicit-differentiation-based derivation can lead to a further computational gain in large-scale meta learning. In Appendix E, we investigate the effect of this identity approximation in the "biased regression" setting where the closed-form solution can be analytically calculated, and empirically show that the identity approximation still allows for accurate estimation of the meta gradient $\frac{\partial L_{meta}}{\partial \lambda}$ and the optimal meta solution $\lambda^*$, even when the true base Jacobian is not an identity matrix.

## 3.2 Algorithmic Adaptation for Adaptive Optimizers

**Problem**   Most existing implicit differentiation algorithms [21, 26, 40] compute the best-response Jacobian in Eq. (1) based on the assumption that the iterative solver $u$ for base optimization is vanilla SGD, whereas recent large models, exemplified by Transformers [5, 62], are by default trained with adaptive optimizers, such as Adam [32]. While the fixed point condition can be theoretically identical for any gradient-based optimizer at convergence (*i.e.*, $\frac{\partial \mathcal{L}_{base}}{\partial \theta^*} = 0$), researchers in practice approximate $\theta^*$ with a small number of gradient steps, at which the above fixed point condition would unlikely hold. Thus, the inconsistency between assumed and actual optimizers results in an incorrect meta gradient, which is a source of training instabilities and reduced performance in meta learning (as we demonstrate in Table 1).

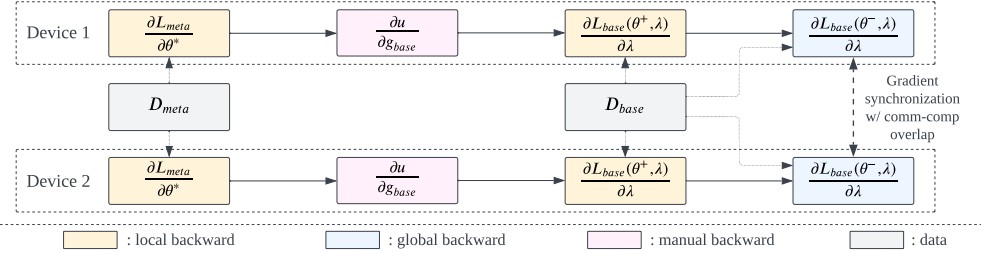

Figure 2: The overall workflow of meta gradient computation with SAMA in the distributed data parallel setting. In detail, SAMA consists of three *first-order* backward passes performed with the underlying automatic differentiation engine, and one manual backward pass for algorithmic adaptation for the adaptive optimizer. Gradient synchronization is performed only once in the last backward pass with communication-computation overlap to minimize the communication bottleneck.

**Solution**  To take adaptive update rules into account, we propose to further expand a meta Jacobian term in Eq. (3) with the chain rule as follows:

$$\frac{\partial u}{\partial \lambda} = \frac{\partial g_{base}}{\partial \lambda} \cdot \frac{\partial u}{\partial g_{base}} = \frac{\partial^2 L_{base}}{\partial \lambda \partial \theta^*} \cdot \frac{\partial u}{\partial g_{base}}$$

$$\implies \frac{\partial L_{meta}}{\partial \lambda} \approx -\frac{\partial u}{\partial \lambda} \cdot \frac{\partial L_{meta}}{\partial \theta^*} = -\frac{\partial^2 L_{base}}{\partial \lambda \partial \theta^*} \cdot \frac{\partial u}{\partial g_{base}} \cdot \frac{\partial L_{meta}}{\partial \theta^*} \tag{4}$$

where $g_{base}$ is base-gradient computed at convergence (*i.e.* $\frac{\partial L_{base}}{\partial \theta^*}$). In short, we accomplish the algorithmic adaptation for any adaptive optimizer with the update rule $u$ through the middle term $\frac{\partial u}{\partial g_{base}}$ in Eq. (4), which reduces to an identity matrix in the case of SGD. To analyze the adaptation cost, we note that parameter updates $u$ in most optimizers are performed only with element-wise operations. Thus, the adaptation matrix is diagonal, for which computation/memory complexities are only $\mathcal{O}(n_b)$. As a concrete example, we provide an adaptation matrix for the most popular Adam optimizer [32] in Appendix C.

Furthermore, to reduce computation/memory complexities involving the costly second-order derivative $\frac{\partial^2 L_{base}}{\partial \lambda \partial \theta^*}$, we instead perform the matrix-vector product with the central-difference method from DARTS [38] and the associative rule of matrix multiplication. All combined, we propose SAMA, a highly compute/memory efficient meta gradient algorithm for scalable meta learning, the formulation of which is as follows:

$$\frac{\partial L_{meta}}{\partial \lambda} \approx -\frac{\partial^2 L_{base}}{\partial \lambda \partial \theta^*} \cdot \left( \frac{\partial u}{\partial g_{base}} \cdot \frac{\partial L_{meta}}{\partial \theta^*} \right) \approx -\frac{\frac{\partial L_{base}(\theta^+, \lambda)}{\partial \lambda} - \frac{\partial L_{base}(\theta^-, \lambda)}{\partial \lambda}}{2\epsilon} \tag{5}$$

where $\theta^{\pm} = \theta^* \pm \epsilon v$ with the perturbation vector $v = \frac{\partial u}{\partial g_{base}} \cdot \frac{\partial L_{meta}}{\partial \theta^*}$ and the step size $\epsilon = \frac{\alpha}{\|v\|_2}$. We empirically observe that $\alpha = 1.0$ works well across a multitude of tasks without further tuning. Finally, we notice that prevoius work in penalty-based bilevel optimization (*e.g.* F2SA [34] and BOME [37]) further uses the direct gradient $\frac{\partial L_{meta}}{\partial \theta^*}$ explicitly in the base level optimization to maximize the performance of the final base parameter $\theta^*$ on the meta objective. Given that our perturbation vector $v$ includes the direct gradient term, we also follow a similar strategy and update the base parameter $\theta$ in the direction of $v$ (*i.e.* $\theta_{t+1} = \theta_t - \epsilon v$) every time the meta update is performed.

### 3.3 Efficient Distributed Training & Implementation

**Problem**  In large-scale learning, distributed data parallelism (DDP), which communicates and synchronizes local gradients from each device before the parameter update, is necessary to improve both compute and memory efficiency. However, most automatic differentiation libraries like Py-Torch [46] only have native DDP support for their basic backward function, whereas SAMA, similar to other meta gradient algorithms, requires a custom implementation of the backward function, which consists of three basic backward passes as shown in Eq. (5).[2] In addition, while a few meta learning

---

[2]Eq. (5) technically involves four derivatives in total, but the adaptation matrix $\frac{\partial u}{\partial g_{base}}$ can be calculated analytically without backpropagation.

libraries, such as Betty [6], have preliminary DDP support, they do not offer further communication cost optimization. As a result, meta learning research to date has either been limited to a single-GPU setup or suffered from communication inefficiency.

**Solution** Since we avoid any explicit computations of second-order gradient information in SAMA, we can utilize various efficient distributed training tricks that have been implemented for first-order gradients. More specifically, to enable efficient DDP in SAMA, we develop a novel communication strategy that performs first two backward passes locally on each device, and then overlaps computation in the final backward pass with communication. In PyTorch, this can be neatly achieved by implementing the first two backward passes with `torch.autograd.grad` and the last one with `torch.autograd.backward`. The overall workflow diagram of SAMA is presented in Figure 2. To facilitate research in scalable meta learning, we provide our implementation of SAMA with the above communication optimization in Betty[3] that only requires a one-line change in the configuration.

## 4 Experiments

While few-shot learning has traditionally been the most popular application of meta learning, most recent large models such as GPT [5], ViT [13], and Whisper [50], provide few-shot generalization capability out of the box. Therefore, we in this work focus on another promising application of meta learning, *data optimization*, where we transform (*e.g.*, reweight, label correct) downstream/pretraining data given the specific objectives in the meta optimization problem. Indeed, there is an increasing number of works originating in data-centric AI that empirically show that the quality of training data significantly affects the final performance of large models [1, 5, 18, 50, 56, 65]. Nevertheless, solutions proposed in these works to improve training data quality mostly rely on hand-designed heuristics, which typically result in suboptimal performance. Given that training data of large models serves as extremely high-dimensional inductive biases in machine learning, we expect that GBML's ability to efficiently optimize high-dimensional inductive biases can be fully unlocked in the data optimization application.

From a technical perspective, large-scale data optimization has a substantial compute/memory cost and frequently involves models that are trained with adaptive optimizers. Therefore, it serves as an ideal benchmark to (a) evaluate the scalability of SAMA compared to existing approaches, (b) study the effectiveness of each component in SAMA, and (c) investigate the practical usefulness of scalable meta learning across diverse domains. Specifically, in this section, we consider three data optimization applications, namely: Noisy finetuning of large language models (Sec. 4.1), continued pretraining of large language models (Sec. 4.2), and scale-agnostic data pruning (Sec. 4.3). While not an application of data optimization, we also include a preliminary analysis on the effect of model size on few-shot image classification accuracy in Appendix D. Finally, we note that all experiment details including baselines, hyperparameters, and compute resources are provided in Appendix B.

### 4.1 Noisy Finetuning of Large Language Models

Weak supervision [53] proposes to achieve a significant reduction in the data labeling cost by letting users quickly generate labels for a large amount of data by exploiting multiple weak labeling functions, such as hand-designed rules and other neural networks. While increasing the *quantity* of labeled data with weak supervision has led to noticeable improvements in multiple applications, a poor *quality* of generated labels results in a degradation in test accuracy, leaving room for further improvement. Here, we attempt to alleviate this data quality issue in weak supervision by utilizing meta learning to automatically optimize noisy training data guided by a small amount of clean data in the meta level. In detail, we use data reweighting [58] and label correction [70] as our data optimization operations, of which the bilevel optimization formulation is as follows:

$$\lambda^* = (\lambda_r^*, \lambda_c^*) = \operatorname*{argmin}_{\lambda_r, \lambda_c} \mathcal{L}(D_{clean}; \theta^*(\lambda_r, \lambda_c))$$

$$\text{s.t. } \theta^*(\lambda_r, \lambda_c) = \operatorname*{argmin}_{\theta} \frac{1}{|D_{noisy}|} \sum_{(x,y) \in D_{noisy}} w(\mathcal{L}; \lambda_r) \cdot \mathcal{L}(f(x; \theta), c(x, y; \lambda_c))$$

where $w(\cdot; \lambda_r)$ and $c(\cdot; \lambda_c)$ are meta learners, respectively, for data reweighting and label correction. To evaluate the scaling efficiency of SAMA, we perform text classification with a BERT-base model

---

[3]`https://github.com/leopard-ai/betty`

with 110M parameters on multiple weak supervision datasets from the WRENCH benchmark [67]. Furthermore, to study the effectiveness of our algorithmic adaptation strategy (Sec. 3.2), we conduct experiments with a variant of SAMA (*i.e.*, SAMA-NA) that does not include algorithmic adaptation, and present the experiment results in Table 1.

| | Algorithm | TREC | SemEval | IMDB | ChemProt | AGNews | Yelp |
|---|---|---|---|---|---|---|---|
| Finetune (orig) [67] | - | 66.56 (2.31) | 83.93 (1.74) | 79.73 (2.60) | 56.09 (1.08) | 86.27 (0.53) | 82.26 (3.50) |
| COSINE [66] | - | 76.56 (0.08) | 86.80 (0.46) | 82.98 (0.05) | 58.47 (0.08) | 87.03 (0.00) | 89.22 (0.05) |
| Finetune (ours) | - | 67.93 (2.55) | 79.28 (1.78) | 78.16 (2.28) | 57.35 (1.43) | 85.79 (0.49) | 84.32 (2.55) |
| +R | SAMA-NA | 74.33 (2.34) | 87.11 (1.11) | 81.92 (1.74) | 60.88 (0.60) | 86.83 (0.19) | 80.96 (3.04) |
| +R & C | SAMA-NA | 79.00 (2.62) | 87.67 (1.36) | 80.44 (0.97) | 64.05 (0.52) | 87.05 (0.39) | 80.73 (3.11) |
| +R | SAMA | 85.73 (0.81) | **89.67** (0.67) | 84.31 (1.86) | 76.89 (1.39) | 89.05 (0.34) | 93.64 (0.40) |
| +R & C | SAMA | **87.93** (1.17) | 88.83 (2.32) | **85.71** (0.82) | **77.78** (0.59) | **89.79** (0.27) | **93.77** (0.08) |

Table 1: WRENCH results. R and C in the first column stand for data reweighting and label correction operations. The number in parentheses indicates standard deviation for each experiment over 3 runs.

In this experiment, we are able to make two observations. First, we notice that SAMA-based data reweighting and label correction both lead to noticeable improvements over finetuning and self-training (COSINE [66]) baselines in noisy text classification accuracy of the *large* Transformer model across all benchmarks with the help of small additional clean data $D_{clean}$ in the meta level. Second, given the superior performance of SAMA compared to SAMA-NA, we empirically verify the importance of algorithmic adaptation when an adaptive optimizer is used to train this Transformer base learner.

Additionally, we compare compute/memory efficiency of SAMA to that of two other implicit-differentiation-based meta learning algorithms, namely Neumann Series [40] and conjugate gradient [51]. For fair comparison, we evaluate GPU memory usage (MB) and throughput (samples/sec) on the AGNews dataset from Wrench with a fixed *global* batch size of 48, and summarize the result in Table 2. Three observations that can be made here are that (1) SAMA is generally more compute/memory efficient than the baselines, (2) the cost of algorithmic adaptation for adaptive optimizers is marginal as expected, and (3) the efficiency

| | GPUs | Memory | Throughput |
|---|---|---|---|
| Neumann | 1 | 26.0 | 82.9 |
| CG | 1 | 28.4 | 82.1 |
| SAMA-NA | 1 | 13.7 | 144.1 |
| SAMA | 1 | 14.3 | 142.0 |
| SAMA | 2 | 10.4 | 241.2 |
| SAMA | 4 | 7.4 | 396.7 |

Table 2: Memory and throughput analysis on AGNews with 4 V100 GPUs.

gap further widens as we distribute compute/memory across multiple GPUs with our efficient DDP communication strategy. Since we were only able to conduct experiments with up to 4 V100 GPUs due to the limited compute resources, exploring extremely large-scale GBML with larger GPU servers remains a promising future research direction. In Appendix F, we provide a more extensive ablation study for each component of SAMA, as well as compare test accuracy, GPU memory usage, and throughput of SAMA against various meta learning algorithms on IMDB and AGNews datasets from the Wrench benchmark.

## 4.2 Continued Pretraining of Large Language Models

DAPT/TAPT [24] empirically demonstrate that additional pretraining (*i.e.*, continued pretraining) of the generic language model on the domain or task-specific data can further improve downstream performance on diverse benchmarks. However, the inclusion of low-quality samples for continued pertaining tasks can potentially hinder pretraining by amplifying negative interference [64], which could lead to suboptimal downstream performance. Here, we attempt to minimize such negative transfer by reweighting samples from the continued pretraining task with meta learning. To this end, we adopt the auxiliary learning technique from TARTAN [11] and simplify the two-stage pretraining-finetuning pipeline into a one-stage multitask learning pipeline with the reweighting scheme applied to the pretraining loss. The bilevel optimization formulation is as follows:

$$\lambda^* = \operatorname*{argmin}_{\lambda} \mathcal{L}_{ft}(D_{ft}; \theta^*(\lambda))$$

$$\text{s.t. } \theta^*(\lambda) = \operatorname*{argmin}_{\theta} \mathcal{L}_{ft}(D_{ft}; \theta) + \frac{1}{|D_{pt}|} \sum_{x \in D_{pt}} w(x; \lambda) \cdot \mathcal{L}_{pt}(x; \theta)$$

where $\mathcal{L}_{ft}/\mathcal{L}_{ft}$ are finetuning/pretraining loss functions, $D_{ft}/D_{pt}$ are finetuning/pretraining datasets, and $w(\cdot; \lambda)$ is the data reweighting network. Following the experiment setup in TARTAN [11], we use task-specific data and a masked language modeling loss in our auxiliary task and perform experiments with RoBERTa-base on 4 datasets from the original DAPT/TAPT paper. We compare our SAMA-based data optimization against DAPT and TARTAN-MT. We exclude TAPT and TARTAN-Meta respectively because (1) TAPT consistently underperforms TARTAN-MT [9] and (2) TARTAN-Meta uses additional validation data in the meta level of the downstream tasks, making the comparison unfair. We report our experiment results in Table 3.

|  | ChemProt | HyperPartisan | ACL-ARC | SciERC | Average |
|---|---|---|---|---|---|
| Baseline | 82.70 (0.45) | 89.03 (2.25) | 68.17 (2.52) | 79.83 (0.89) | 79.93 |
| DAPT [24] | 84.17 (0.50) | 87.23 (3.65) | 71.84 (4.78) | 80.42 (1.57) | 80.92 |
| TARTAN-MT [11] | 84.18 (0.30) | 94.64 (0.91) | **72.41** (1.94) | 80.83 (0.71) | 83.02 |
| SAMA (ours) | **84.49** (0.13) | **95.18** (0.03) | 71.63 (1.68) | **81.84** (0.08) | **83.29** |

Table 3: Experiment results for auxiliary learning with the continued pretraining task. Following [24], we report test micro-F1 for ChemProt and macro-F1 for the other datasets. The number in parentheses indicates the standard deviation for each experiment over 3 runs.

As shown above, SAMA-based data optimization leads to improvements in downstream performance on almost all datasets. This indirectly demonstrates that SAMA-based data reweighting can identify more/less relevant data in the auxiliary task and accordingly up-/down-weight them, unlike TARTAN-MT which allocates equal importance weights on all auxiliary data. Therefore, we expect that our method would likely benefit from additional auxiliary data by automatically figuring out and exploiting only relevant data, whereas TARTAN-MT is much more susceptible to negative transfer. While we only used task-specific data in our auxiliary task for the fair comparison with TARTAN-MT, extending auxiliary data to domain-specific or even general text data and comparing SAMA against DAPT or TARTAN-MT would be an intriguing future research direction. Finally, we analyze the GPU memory usage of different-sized RoBERTa in this experiment and present the result in Figure 1. The figure clearly shows the superior memory efficiency of SAMA with the increasing model size.

### 4.3 Scale-Agnostic Efficient Data Pruning

Data pruning [47, 59, 61, 63] has recently received the limelight in the machine learning community as a means to both improve training efficiency and reduce (semantic) redundancy in training data. In particular, Sorscher et al. [59] showed both theoretically and experimentally that neural scaling laws can be beaten by data pruning. Nevertheless, they point out that the optimal data pruning metric varies across different dataset scales and further research in scalable data pruning metrics is needed. Here, we propose to forgo hand-designed data pruning metrics, and rather automatically *meta-learn* the importance weight of each training data following Meta-Weight-Net (MWN) [58] with four major modifications. First, we replace their iterative differentiation meta gradient algorithm with SAMA to achieve improved memory/compute efficiencies. Second, we further speed up meta learning by enabling distributed training with our efficient communication strategy. Third, we use the uncertainty of the prediction in addition to the loss value as an input to MWN to better estimate importance weight of each training data. Last, we use training data *both* in the base and the meta levels, assuming no additional validation data. A bilevel optimization formulation of our method is as follows:

$$\lambda^* = \underset{\lambda}{\operatorname{argmin}} \, \mathcal{L}(D_{train}; \theta^*(\lambda))$$

$$\text{s.t. } \theta^*(\lambda) = \underset{\theta}{\operatorname{argmin}} \, \frac{1}{|D_{train}|} \sum_{(x,y) \in D_{train}} w(\mathcal{L}, \mathcal{U}; \lambda) \cdot \mathcal{L}(x, y; \theta)$$

where $w(\cdot; \lambda)$ is MWN that takes the loss value $\mathcal{L}$ and the uncertainty $\mathcal{U}$ of the training sample $(x, y)$ as an input and outputs the importance weight. Under this setup, we run meta learning with SAMA for 30 / 50 epochs respectively for ImageNet-1k / CIFAR-10 and obtain the pruning metrics by averaging the importance weights of the last 5 epochs. We compare our method to several popular static/dynamic data pruning baselines, and present the results in Figure 3.

As expected, GBML-based data pruning with SAMA not only outperforms heuristics-based data pruning but also works well across different dataset scales. Surprisingly, we observe that GBML-based data pruning even leads to improvements in test accuracy at the pruning ratio of 0.1 and 0.2

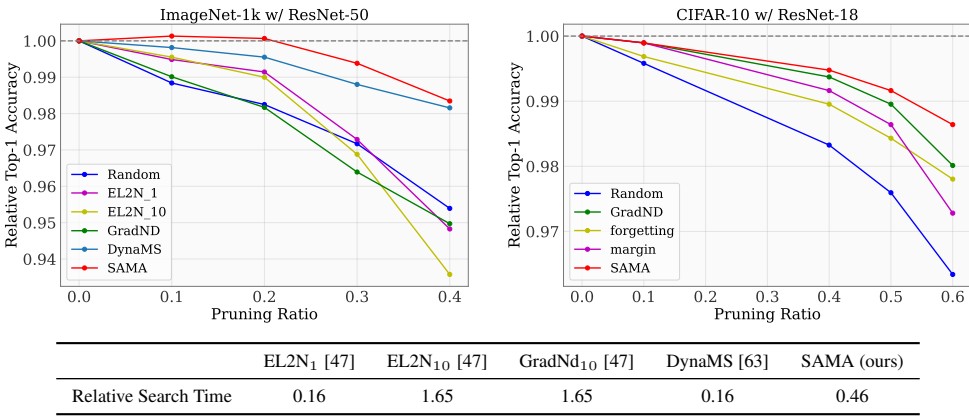

| | EL2N$_1$ [47] | EL2N$_{10}$ [47] | GradNd$_{10}$ [47] | DynaMS [63] | SAMA (ours) |
|---|---|---|---|---|---|
| Relative Search Time | 0.16 | 1.65 | 1.65 | 0.16 | 0.46 |

Figure 3: **Top Left:** ImageNet-1k data pruning results with ResNet-50. Reported numbers are relative accuracy compared to full training accuracy (*i.e.*, pruned_acc/full_acc). Accuracy for other baseline methods is obtained from DynaMS [63]. **Top Right:** CIFAR-10 data pruning results with ResNet-18. Accuracy for other baseline methods is obtained from Deepcore [22]. **Bottom:** Relative time spent in finding data to prune compared to full ImageNet-1k training time.

on ImageNet-1k. The potential implication is that ImageNet-1k may have noisy labels or semantic redundancy and that GBML is able to automatically figure and filter out these samples. Further in-depth investigation of filtered data remains an interesting research direction. Considering that compute/memory inefficiency has traditionally been the major bottleneck in GBML applications, we also compare the relative search time for data pruning. Our result shows that SAMA demonstrates comparable or even shorter search time than heuristics-based methods. We also note that, while the original MWN [58] encounters the OOM error under our setup of batch_size=256, the throughput analysis with the reduced batch size reveals that efficient distributed training with SAMA on 4 GPUs achieves 15-20× speed up compared to the original MWN that lacks distributed training support.

## 5   Related Work

**Algorithms**   Two major lines of research in gradient-based meta learning algorithms are iterative and implicit differentiation [19]. Iterative differentiation [14, 15, 16, 42] computes meta gradients by differentiating through the optimization path and therefore requires saving all intermediate states on the path. This makes the memory/compute costs of iterative differentiation increase linearly with the number of unrolling steps. While the linear cost can be avoided with the use of techniques like truncated backpropagation [36], it is still more expensive than that of implicit differentiation, in which meta gradient computation is independent of the length of the optimization path. More specifically, meta gradient computation in implicit differentiation depends *only* on the final state of the optimization path. To compute base Jacobian inversion in implicit differentiation, a multitude of variants have been proposed, each of which uses Neumann series [7, 40], conjugate gradient [51], Nystrom method [25], and more. While generally being more compute/memory efficient than iterative differentiation, most existing implicit differentiation algorithms have poor scalability due to the issues studied in Sec. 3.

**Applications**   Meta learning has found many applications in machine learning including few-shot learning [14, 51, 71], neural architecture search [38, 69], hyperparameter optimization [15, 16, 40, 42], data optimization [21, 54, 58, 70], and reinforcement learning [23, 29, 52], to name a few. Notably, most of these applications share the underlying mathematical formulation of bilevel optimization and are conceptually related to optimal design and inductive/declarative programming paradigms.

**Systems**   Compared to algorithms and systems research, there are relatively fewer research efforts in meta learning systems. In an attempt to facilitate research in few-shot image classification, *higher* [20], *learn2learn* [3], and *TorchMeta* [9] have been developed. However, due to their specific focus on few-shot image classification, these libraries have not been actively used in other meta learning tasks, such as data optimization or neural architecture search. Recently, software libraries

for implicit differentiation including *JaxOpt* [4] and *Betty* [6], have been proposed. Given that Betty's software architecture is specifically designed to support various systems optimization for large-scale meta learning, we chose to implement SAMA in this framework.

## 6    Conclusion

In this paper, we strived to make scalable meta learning practical via both algorithmic and systems advancements. Towards this goal, we investigated diverse scaling bottlenecks in meta learning at a technical level and resolved them by developing SAMA. Tested on multiple benchmarks, SAMA empirically demonstrated its scaling efficiency as well as its capability to optimize a variety of high-dimensional inductive biases of large-scale learning. In future work, we plan to explore two directions. First, given that training extremely large models with 10B+ parameters require various systems techniques such as model/pipeline parallelism or optimizer sharding, extending SAMA to be compatible with these techniques would be highly important for further scalability. Second, we will also focus on large-scale meta learning application research, such as neural architecture search for Transformer-family models. Overall, we hope that our work can serve as a stepping stone for a lot of interesting scalable meta learning research to come.

**Limitations & Broader Impacts**    While SAMA has demonstrated significantly improved compute/memory efficiencies, and stably worked with fairly large models like BERT/RoBERTa, we could not test it on larger models with 1B+ parameters due to a lack of computational resources. Meta learning itself is mostly value-neutral, but there is a chance that practitioners amplify the bias or toxicity of the machine learning programs by enforcing them in the meta objective.

## Acknowledgements

We thank all the reviewers for their invaluable comments and feedback. We like to acknowledge CMU Workhorse and TIR group for providing compute resources for this work. EX and SKC acknowledge the support of NSF IIS2311990, NSF IIS2123952, NSF CNS2008248, NSF BCS2040381, NGA HM04762010002, NIGMS R01GM140467, and Amazon 2023-5007107. WN was supported in part by NSF (#1651565), AFOSR (FA95501910024), ARO (W911NF-21-1-0125), CZ Biohub, and Sloan Fellowship. ES and SVM acknowledge the support in part by DSO National Laboratories.

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

# A  Philosophies behind SAMA & Scalable Meta Learning

Here, we additionally discuss several important design principles and philosophies behind scalable meta learning and SAMA in a Q&A format.

**Q. Why do we study scalable meta learning?**

**A.** Richard Sutton points out in his article "The Bitter Lesson" [60] that machine learning algorithms that stand the test of time are ones that continue to *scale gracefully with the increased computation budget* (*i.e.*, scalable algorithms). Given that meta learning is an important topic in machine learning with many applications including data optimization [54], hyperparameter optimization [16], few-shot learning [14, 51], and adversarial learning [45], it was a natural call for us to investigate the scalability of meta learning algorithms following the spirit of "The Bitter Lesson". Interestingly, such a focus on the scalability of meta learning algorithms distinguishes our work from most other meta learning works, in which the typical focus is to improve the overall performance of meta learning algorithms *under a limited computation budget* (usually bounded by a single GPU).

**Q. What are the design principles behind scalable meta learning?**

**A.** The increased computation budget powered by hardware advancements (*e.g.*, Moore's law) has evolved a new ecosystem of large models and datasets in machine learning over time, which involves both systems and algorithms components. For example, to efficiently leverage the increased computation for large-scale learning, diverse systems techniques, such as data/model/pipeline parallelism have been developed [28, 35, 57]. At the same time, researchers have devised various algorithms that are highly effective for large-scale learning, such as backpropagation [55], skip connections [27], Adam optimizer [32], self-attention [62], etc. Accordingly, in addition to guaranteeing memory/compute efficiency for scalability, our major design principle for scalable meta learning was to *ensure compatibility with existing systems and algorithms* in the large-scale learning ecosystem.

**Systems compatibility**  Given that a great deal of systems support in machine learning, such as communication-computation overlap [35], has been developed for first-order gradient methods, avoiding explicit computations of higher-order gradient information including Hessian-vector products was an important design principle in SAMA. Even though we mostly explored distributed training in this work, SAMA is also compatible with other system features such as half-precision training and activation checkpointing, which could further improve memory efficiency.

**Algorithms compatibility**  While there exist several meta learning algorithms that avoid the computation of higher-order gradient information [34, 38, 37], many of these algorithms either assume the use of a naive SGD update rule or devise specific update rules tailored to their own algorithms at the base level, significantly hampering their algorithm compatibility. In contrast, SAMA allows for the use of arbitrary optimizers at the base level via algorithmic adaptation.

# B Experiment Details

In this section, we discuss various experiment details such as hyperparameters, baselines, and compute resources used for our experiments in Section 4. Our experiment codes are available in the supplementary material as well as under the official Betty GitHub repository [4].

## B.1 Noisy Finetuning of Large Language Models

**Hyperparameters** We ran training for 1000 iterations on TREC/SemEval/IMDB/ChemProt/Yelp/AGNews datasets from the WRENCH benchmark [67], with a batch size of 32, a weak supervision algorithm of majority voting, and the hyperparameters in Table 4 below.

|  | model | optimizer | init_lr | lr_scheduler | wdecay | dataset | unroll step | SAMA $\alpha$ |
|---|---|---|---|---|---|---|---|---|
| Base | BERT-base | Adam | 1e-5 | cosine | 0 | WRENCH train set (with majority voting) | 10 | 1.0 |
| Meta (Reweight) | 2-layer MLP | Adam | 1e-5 | None | 0 | WRENCH dev set | N/A | N/A |
| Meta (Correct) | 2-layer MLP | Adam | 1e-5 | None | 0 | WRENCH dev set | N/A | N/A |

Table 4: Hyperparameters for *noisy finetuning of large language models* experiments.

**Baselines** We adopted naive finetuning and self-training (*i.e.*, COSINE [66]) approaches from the original WRENCH benchmark paper [67] as our baseline.

**Compute Resources** We used 1 NVIDIA RTX 2080Ti GPU for the main experiment, and 4 NVIDIA Tesla V100 GPUs for the throughput-memory analysis in Table 2 and Figure 1.

## B.2 Continued Pretraining of Large Language Models

**Hyperparameters** We ran training for 100 epochs with a batch size of 16, a maximum sequence length of 256, and the hyperparameters in Table 5 below.

|  | model | optimizer | init_lr | lr_scheduler | wdecay | dataset | unroll step | SAMA $\alpha$ |
|---|---|---|---|---|---|---|---|---|
| Base (Downstream) | RoBERTa-base | Adam | 2e-5 | linear decay + warmup linear (warmup proportion 0.6) | 0 | train split of ChemProt/HyperPartisan/ ACL-ARC/SciERC | 10 | 0.3 |
| Base (Auxiliary) | RoBERTa-base | Adam | 2e-5 | linear decay + warmup linear (warmup proportion 0.6) | 0 | train split of ChemProt/HyperPartisan/ ACL-ARC/SciERC | 10 | 0.3 |
| Meta | 2-layer MLP | Adam | 1e-5 | None | 0 | train split of ChemProt/HyperPartisan/ ACL-ARC/SciERC | N/A | N/A |

Table 5: Hyperparameters for *continued pretraining of large language models* experiments.

**Baselines** We adopt DAPT [24] and TARTAN-MT [11] as our baselines for this experiment. In detail, DAPT [24] performs additional masked language model pretraining on domain-specific data on top of the pretrained RoBERTa-base model and then finetunes the model on the downstream text classification task. We follow [24] (see Table 14 in the original paper) for setting downstream finetuning hyperparameters. Alternatively, TARTAN-MT [11] performs masked language modeling with task specific data and downstream text classification training simultaneously in a multitask fashion through two different heads.

**Compute Resources** We used 1 NVIDIA Tesla V100 GPU for the main experiment, and 1 NVIDIA RTX A6000 GPU for the "memory vs model-size analysis" in Figure 1.

## B.3 Scale-Agnostic Efficient Data Pruning

**Hyperparameters** We ran meta learning for 30 epochs with a batch size of 256 and the configuration shown in Table 6 below. After pruning data based on the meta learning result, we ran ImageNet-1k

---

[4]`https://github.com/leopard-ai/betty/tree/main/examples`

training for 120 epochs with the learning rate decayed by 10 at epochs [40, 80] following the set up in DynaMS [63].

| | model | optimizer | init_lr | lr_scheduler | wdecay | dataset | unroll step | SAMA $\alpha$ |
|---|---|---|---|---|---|---|---|---|
| Base | ResNet-50 | SGD | 1e-1 | None | 1e-4 | ImageNet-1k train set | 2 | 1.0 |
| Meta | 2-layer MLP | Adam | 1e-5 | None | 0 | ImageNet-1k train set | N/A | N/A |

Table 6: Hyperparameters for *ImageNet-1k data pruning* experiments

For the CIFAR-10 data pruning experiment, we ran meta learning for 50 epochs with a batch size of 128, and configuration in Table 7 below. After pruning the data based on the meta learning result, we ran CIFAR-10 training for 200 epochs with the cosine learning rate decay schedule following the setup in DeepCore [22].

| | model | optimizer | init_lr | lr_scheduler | wdecay | dataset | unroll step | SAMA $\alpha$ |
|---|---|---|---|---|---|---|---|---|
| Base | ResNet-18 | SGD | 1e-1 | None | 5e-4 | CIFAR-10 train set | 2 | 1.0 |
| Meta | 2-layer MLP | Adam | 1e-5 | None | 0 | CIFAR-10 train set | N/A | N/A |

Table 7: Hyperparameters for *CIFAR-10 data pruning* experiments

**Baselines** We adopt EL2N [47], GraNd [47], DynaMS [63] as our baselines for the ImageNet-1k experiments and GraNd [47], forgetting [61], margin [8] for the CIFAR-10 experiments. In detail, EL2N/GraND [47] respectively select samples with large L2-loss/gradient-norm values, forgetting [47] chooses samples that are frequently forgotten during training, and margin [8] chooses samples with least confidence. While these baselines are considered *static pruning*, DynaMS [63] falls under the category of dynamic pruning where data to be pruned change during training. Dynamic pruning may see the whole training data across different epochs, making a fair comparison difficult. Surprisingly, despite being a static pruning algorithm, SAMA-based data pruning still achieves a better performance than DynaMS.

**Compute Resources** We used 4 NVIDIA Tesla V100 GPUs for Imagenet-1k data pruning meta learning experiments and 1 NVIDIA RTX 2080Ti GPU for CIFAR-10 experiments.

**Additional Information** We measured the uncertainty $\mathcal{U}$ via the difference between the predictions of the current model and the exponentially-moving-averaged model.

## C   Algorithmic Adaptation for Adam Optimizer

Since the Adam optimizer [32] has been the most popular optimizer to train large models, exemplified by Transformers [68], here we provide the adaptation matrix for Adam. We denote the first and second moments of the gradient in Adam as $m$ and $v$ respectively, and the learning rate as $\gamma$.

$$
\begin{aligned}
\frac{\partial u_{adam}}{\partial g} &= \frac{\partial u}{\partial g}\left(\gamma\frac{\beta_1 m + (1-\beta_1)g}{\sqrt{\beta_1 v + (1-\beta_1)g^2} + \epsilon}\right) \\
&= \gamma\frac{(1-\beta_1)\beta_2 v - (1-\beta_1)\beta_2 mg + (1-\beta_1)\epsilon\sqrt{\beta_1 v + (1-\beta_1)g^2}}{\sqrt{\beta_1 v + (1-\beta_1)g^2}\left(\sqrt{\beta_1 v + (1-\beta_1)g^2} + e\right)^2} \\
&\approx \gamma\frac{(1-\beta_1)\beta_2 v - (1-\beta_1)\beta_2 mg}{\sqrt{\beta_1 v + (1-\beta_1)g^2}\left(\sqrt{\beta_1 v + (1-\beta_1)g^2} + e\right)^2} \qquad \text{(because } \epsilon \ll 1)
\end{aligned}
$$

Adaptation matrices can be similarly derived for other adaptive optimizers.

## D   The Effect of Scaling in Model-Agnostic Meta Learning

Since the inception of MAML [14], a myriad of algorithms have been proposed to improve few-shot image classification while assuming a fixed network architecture. In contrast, here we shift our focus from the algorithm to the scale, and propose to study the following question: "Leveraging the compute/memory efficiency of SAMA, can we improve the few-shot generalization capability by scaling up the network size?". Since SAMA is a variant of implicit differentiation, we closely follow the experiment setup in iMAML [51], where proximity to the initialization weights is explicitly enforced by $L_2$-regularization. The major difference is that iMAML uses a conjugate-gradient-based method, which requires second-order gradient information to compute meta gradients, while we adopt SAMA to achieve improved scaling to larger networks with its superior memory/compute efficiency. We conduct preliminary experiments on the Omniglot 20-way 1-/5-shot tasks with the basic 4-layer CNN architecture, while varying the width (hidden size) of the networks to study the effect of the model size on the few-shot classification accuracy. The experiment results are provided in Figure 4 below.

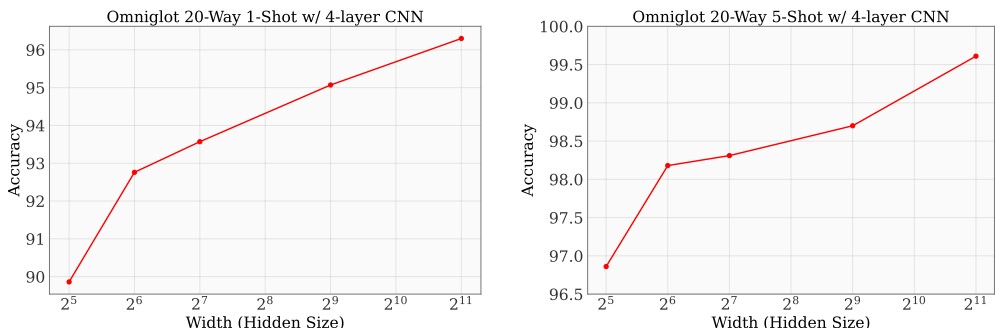

Figure 4: Few-shot image classification accuracy on Omniglot 20-way 1-/5-shot tasks with varying network sizes.

Interestingly, we observe that the increased model size leads to consistent improvements in few-shot classification accuracy. The important question following this observation is "can we apply scaling laws [30] from other tasks (*e.g.*, language modeling) to general meta learning beyond few-shot image classification?" Since meta learning involves two optimization problems (meta and base) unlike traditional machine learning problems, it is as of now unclear how to define the general concept of "scale" in terms of both model and dataset sizes. We expect that further research in this direction would be critical in systematically studying scalable meta learning.

# E    Justification of the Identity Approximation for Base Jacobian

In this section, we aim to study the empirical effect of our identity approximation for base Jacobian on the meta gradient $\frac{\partial L_{meta}}{\partial \lambda}$ and the optimal meta solution $\lambda^*$, when the true base Jacobian is not an identity matrix. As obtaining the closed-form solution of the Hessian is impossible in almost all deep learning problems, we study the soundness of the identity approximation of base Jacobian in the simpler "biased regression" setting [19], for which the bilevel optimization formulation is as follows:

$$\lambda^* = \arg\min_{\lambda} \|X'w^*(\lambda) - y'\|^2$$

$$w^*(\lambda) = \arg\min_{w} \|Xw - y\|^2 + \beta\|w - \lambda\|^2$$

Given the above formulation, the closed-form solutions for the base Jacobian, the meta-gradient $g_\lambda$, and the optimal meta solution $\lambda^*$ are:

1. Base Jacobian $= X^TX + \beta I$
2. $g_\lambda = \beta(X^TX + \beta I)^{-1}(X'^TX'w^* - X'^Ty')$, where $w^* = (X^TX + \beta I)^{-1}(X^Ty + \beta\lambda)$
3. $\lambda^* = (A^TA)^{-1}A^Tb$, where $A = \beta X'(X^TX + \beta I)^{-1}$, $b = y' - X'(X^TX + \beta I)^{-1}X^Ty$

We set $\beta = 0.1^5$ and perform 100 meta updates, and measure 1) the cosine similarity between the ground truth $g_\lambda$ and the meta gradient obtained with our approximation (i.e. $g_{SAMA}$), and 2) the L2 distance between the current meta parameter $\lambda_t$ and the optimal solution $\lambda^*$ at each time step $t$. For a more thorough analysis, we also compute these two metrics for other meta gradient algorithms that explicitly approximate base Jacobian inverse with conjugate gradient and Neumann series. In Figure 5, we provide the metric obtained from all time steps.

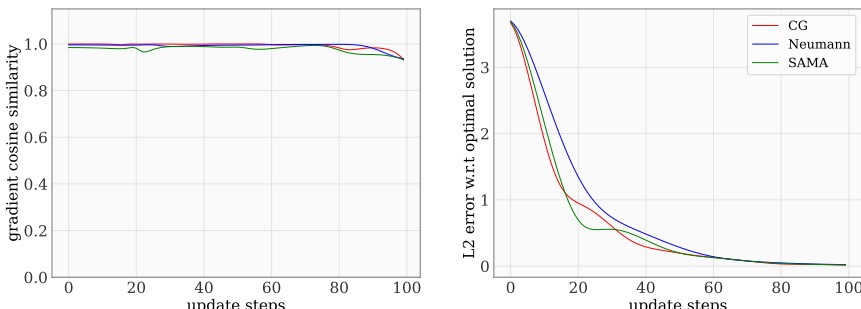

Figure 5: **Left:** $\cos(g_\lambda, g_{approx})$, where $g_\lambda$ is a ground truth (i.e. closed-form) meta gradient, and $g_{approx}$ is an approximate meta gradient obtained with various popular meta learning algorithms including SAMA. **Right:** $\|\lambda_t - \lambda^*\|_2$, where $\lambda_t$ is the meta parameter $\lambda$ after $t$ meta updates.

From Figure 5, it can be clearly seen that 1) while slightly less accurate than second-order algorithms like CG, SAMA still achieves a high directional alignment with the ground truth meta-gradient, and 2) SAMA also achieves a stable convergence to the optimal solution at a comparable speed.

---

[5]We used smaller $\beta$ than in the original paper [19] (1 vs 0.1), to amplify the "non-identitiness" of the base Jacobian.

# F   Extensive Ablation Study

In this section, we perform an extensive ablation study that studies the effectiveness of each component in SAMA (*i.e.* base Jacobian inverse, algorithmic adaptation for the adaptive optimizer, and efficient distributed training) by comparing test accuracy, GPU memory usage, and throughput against various baseline meta learning algorithms. Our ablation study is performed on AGNews and IMDB datasets from the Wrench benchmark [67], and the result is presented below.

| | Base Jacobian | Algo Adapt | Distributed | Accuracy | Throughput | Memory |
|---|---|---|---|---|---|---|
| Finetuning (no meta learning baseline) | x | x | x | 85.79 | 169.16 | 7.77 |
| Iterative Diff (*e.g.* MAML) [14, 42] | x | x | x | 85.78 | 28.07 | 22.94 |
| Conjugate gradient (*e.g.* iMAML) [51] | x | x | x | 86.78 | 65.14 | 22.03 |
| Neumann series [40] | x | x | x | 86.65 | 67.03 | 19.70 |
| DARTS (or $T_1 - T_2$) [38, 41] | o | x | x | 86.36 | 43.69 | 10.81 |
| SAMA-NA | o | x | x | 86.55 | 137.90 | 10.30 |
| SAMA | o | o | x | **89.05** | 134.56 | 11.12 |
| SAMA (2 GPUs) | o | o | o | **88.85** | 226.27 | 8.00 |
| SAMA (4 GPUs) | o | o | o | **89.02** | **298.28** | **6.46** |

Table 8: Ablation results on AGNews

| | Base Jacobian | Algo Adapt | Distributed | Accuracy | Throughput | Memory |
|---|---|---|---|---|---|---|
| Finetuning (no meta learning baseline) | x | x | x | 78.16 | 144.39 | 6.60 |
| Iterative Diff (*e.g.* MAML) [14, 42] | x | x | x | 80.25 | 24.24 | 22.03 |
| Conjugate gradient (*e.g.* iMAML) [51] | x | x | x | 81.01 | 56.27 | 21.92 |
| Neumann series [40] | x | x | x | 79.92 | 57.85 | 19.75 |
| DARTS (or $T_1 - T_2$) [38, 41] | o | x | x | 80.47 | 37.53 | 10.35 |
| SAMA-NA | o | x | x | 81.92 | 117.86 | 9.93 |
| SAMA | o | o | x | **84.31** | 116.94 | 10.84 |
| SAMA (2 GPUs) | o | o | o | **85.18** | 196.48 | 7.84 |
| SAMA (4 GPUs) | o | o | o | **84.19** | **263.74** | **6.39** |

Table 9: Ablation results on IMDB

From our extended ablation study, it can be seen that 1) an identity approximation of base Jacobian significantly improves memory/compute efficiency, 2) algorithmic adaptation improves meta learning performance at the minimal compute/memory cost, and 3) our communication-optimized distributed training further improves compute/memory efficiency.

