# OpenReview forum: "Making Scalable Meta Learning Practical"
_NeurIPS.cc/2023/Conference — NeurIPS 2023 poster_

### Official Review · Reviewer_TSYz · 2023-07-05

**Soundness:** 3 good
**Presentation:** 3 good
**Contribution:** 3 good
**Rating:** 6
**Confidence:** 3

**Summary:**

The paper "Making Scalable Meta Learning Practical" introduces a novel approach called SAMA (Scalable Meta Learning with Arbitrary Optimizers) to address the scalability issues in meta learning. The authors combine advancements in implicit differentiation algorithms and systems to develop SAMA, which supports arbitrary optimizers in meta learning programs while reducing computational burden and utilizing efficient distributed training techniques. In experiments, they focus on data optimization tasks and demonstrate the effect of SMLA on several domains.

**Strengths:**

1 The paper proposes SAMA, a new method that combines implicit differentiation algorithms and systems to address scalability issues in meta-learning. This innovative approach sets it apart from existing methods.

2. The paper presents compelling experimental results, showing significant throughput increases and memory consumption reductions on single- and multi-GPU setups compared to baseline meta-learning algorithms. These performance improvements validate the effectiveness of SAMA.

3. Well written. This paper is well-written and easy to follow.

**Weaknesses:**

-- Need more clarification on the application scenarios.

1. This paper claims that the proposed SAMA makes scalable meta-learning practical while the experiments are only carried out in data optimization applications. However, technically, the method should also be applicable to other applications of meta-learning. In practice, whether the method proposed in this paper is effective for other applications?

2. There are a large number of meta-learning algorithms, some of which may not involve the computation of a large-scale Jacobian matrix. Does the proposed method work for such algorithms? The application scope should be better clarified.

-- Unclear description

3. How is \theta^* obtained? Is it approximated by several steps or does it need to be trained to converge, and if it needs to be trained to converge, does the meta-learning process require more training time? Is such overhead already taken into consideration in the evaluation?

-- Typos and errors

4. There are some spelling errors in this paper that need further correction. Line 259 wrong subscript: the second “ft” should be “pt”. In Equation 5, The second equal sign should also be an approximate equal sign?


**Questions:**

Please see the weakness.

---

> ### Author Rebuttal · Authors · 2023-08-09
>
> We thank you for the positive review as well as the helpful feedback. Here, we address each of the questions and comments that you raise.
>
> ### **Clarification on the application scenarios**
>
> > **Q1.** This paper claims that the proposed SAMA makes scalable meta-learning practical while the experiments are only carried out in data optimization applications. …. In practice, whether the method proposed in this paper is effective for other applications?
>
> **A.** As the reviewer noted, SAMA can be utilized in other meta learning applications that can be formulated as bilevel optimization, such as few-shot learning, neural architecture search, and hyperparameter optimization. As mentioned in line 209, we also performed the traditional (MAML-like) few-shot learning experiments in Appendix D. While most existing few-shot learning works focus on developing new algorithms given the fixed network size, in our experiment we rather study the question of “can we improve the performance of few-shot learning by increasing the model size given the memory/compute efficiency of SAMA?”, to which our preliminary answer is “yes”.
>
> > **Q2.** There are a large number of meta-learning algorithms, some of which may not involve the computation of a large-scale Jacobian matrix. Does the proposed method work for such algorithms?
>
> **A.** We believe several meta-learning algorithms that avoid the use of a large Jacobian matrix (e.g. Reptile [1]) are specifically designed for few-shot learning, which is one particular application of meta-learning/bilevel optimization. Therefore, it is oftentimes not straightforward to apply these algorithms to a more general class of meta learning applications (e.g. data optimization, neural architecture search) that can be formulated as bilevel optimization. In contrast, SAMA is directly derived from the bilevel optimization formulation of meta learning, and thereby demonstrates a more wide applicability.
> As for applying some components of SAMA (e.g. Sec 3.3: communication optimization) to other meta learning algorithms, we believe that it is possible. For example, our communication optimization strategy can be easily transferred to DARTS [2]. However, as each meta learning algorithm has a different design, our components may need to be additionally adapted to the target algorithm.
>
> [1] Nichol et al., On first-order meta-learning algorithms. Arxiv, 2018.
> [2] Liu et al., Darts: Differentiable architecture search. ICLR, 2018.
>
> ### **Unclear description**
>
> > **Q3.** How is $\theta^*$ obtained?
>
> **A.** $\theta^*$ is approximated by several unroll updates instead of by training to convergence. As the reviewer expected, by avoiding full training, we were able to significantly reduce the computation burden (refer to line 97-100). All our experiments are conducted with this several-updates-approximation strategy for a fair comparison. More experiment details including the number of unroll steps are provided in Appendix B.
>
> ### **Typos and errors**
> > **Q4.** There are some spelling errors in this paper that need further correction
>
> **A.** Thanks for pointing out typos. We will update these in the camera ready version of our paper.
>
> We again express our gratitude for your constructive comments, which are very helpful to our paper. If you have any further comments that could make our paper stronger, we are more than happy to discuss them in the remaining review period.

---

### Official Review · Reviewer_YbqA · 2023-07-07

**Soundness:** 2 fair
**Presentation:** 3 good
**Contribution:** 3 good
**Rating:** 6
**Confidence:** 3

**Summary:**

This paper proposes a novel framework that could achieve scalable meta-learning algorithms from the perspectives of algorithms and systems. For algorithms, some approximations are proposed for base Jacobian inverse and adaptive optimizers; for systems, it implements the distributed algorithms to ensure different tasks can be done in parallel. Through extensive experiments in scalable meta-learning experiments, the proposed approach shows advantages over other baselines.

**Strengths:**

1. The motivation is clear and robust. Scalable meta-learning is critical, and this paper might be a good solution.
2. The presentation is clear, although some technical issues are unclear. Generally, the presentation is good.
3. Different experiments regarding scalable meta-learning are conducted, which could be helpful to get some insights for the common meta-learning/few-shot learning researchers.

**Weaknesses:**

1. Although the current experiments are very helpful in exploring scalable experiments under meta-learning, some important ablation study ones are missing.
- 1) Three issues (section 3) are solved to ensure the scalability of meta-learning. Then which issue is the major one to slow down the process? How could these three issues affect the scalabilities? For example, for the DDP, is that possible to conduct some experiments of DDP on the top of MAML? Most researchers are familiar with MAML, and if DDP experiments on MAML show improved efficiency on common few-shot learning experiments, that would be very helpful and easy to understand the benefits. Similarly, for the other two components or issues.
- 2) Is that possible to conduct more experiments to compare the current approach with others, such as iMAML, see Figure 2 in the iMAML paper. It is hard to compare the proposed approach with others based on the current experiments. It is better to compare them based on the published experiments.

   - MAML: Model-Agnostic Meta-Learning for Fast Adaptation of Deep Networks, ICML 2017

   - iMAML: Meta-Learning with Implicit Gradients, NeurIPS 2019

2. It's straightforward for section 3.3, but it is a little bit difficult to follow for sections 3.1 and 3.2.

**Questions:**

1. ablation study experiments (see weakness)
2. how to understand u in section 3.1?
3. data reweighting method in 4.1: what is the relationship between w and $\lambda_r$, c and $\lambda_c$? Are w and c networks? Also, it is helpful to cite similar papers here:
  - Learning to Reweight Examples for Robust Deep Learning, ICML 2018
  - A Nested Bi-level Optimization Framework for Robust Few Shot Learning, AAAI 2022
4. Some typos:
- 1) Eq.5: the negative sign "-" is missing.
- 2) line 259: the second $L_{ft}$ should be ${L}_{pt}$.
- Also, it might be easy to follow if $\lambda$ is included in the first equation below line 69.

**Limitations:**

yes.

---

> ### Author Rebuttal · Authors · 2023-08-09
>
> We thank you for the valuable review. We try our best to address your concerns and questions here and in the **global response**.
>
> ### **Ablation study**
> > **Q.** Three issues (section 3) are solved to ensure the scalability of meta-learning. Which issue is the major one to slow down the process? How do these three issues affect scalability?
>
> **A.** Although we didn’t explicitly frame it as an “ablation study”, the effectiveness of each component in SAMA can be understood from Table 1 & 2 in our paper. Below, we directly discuss each component of SAMA based on Table 1 & 2. We also provide a unified table as an ablation study, in the global response above.
>
> **Base Jacobian inverse**
>
> *tl;dr* Identity approximation of Base Jacobian significantly improves memory/compute efficiency (Table 2).
>
> Our baseline algorithms in Table 2 and Figure 1, Neumann and CG, are both state-of-the-art implicit differentiation meta learning algorithms that attempt to approximate base Jacobian inverse as accurately as possible with multiple Hessian-vector products, instead of approximating it with the identity as SAMA. As shown in Table 2, due to expensive second-order gradient computations involved in Hessian-vector products, these methods demonstrate much poorer throughput and GPU memory usage than SAMA, both of which are the major bottlenecks in efficiently scaling meta learning. In our ablation study in the global response, we additionally demonstrate that this identity approximation also has a minimal impact on accuracy as well by comparing SAMA against Neumann and CG in terms of accuracy.
>
> **Algorithmic adaptation for adaptive optimizer**
>
> *tl;dr* Algorithmic adaptation significantly improves accuracy (Table 1) at the minimal memory/compute cost (Table 2).
>
> The main goal of this work is to devise a *(1) memory/compute efficient* meta learning algorithm that *(2) achieves good performance/accuracy*. In Table 1, SAMA consistently achieves better accuracy than SAMA-NA, which lacks algorithmic adaptation. Moreover, Table 2 shows that SAMA achieves a comparable memory/compute efficiency as SAMA-NA.
>
> **Distributed training**
>
> *tl;dr* Both GPU memory usage and throughput improve consistently as computations are distributed across more GPUs (Table 2).
>
> We agree that it may not be straightforward for readers to clearly understand the benefit of each component when the results are spread across two separate tables. Hence, in the global comment above, we provide a unified table for Wrench experiments, including a comparison with state-of-the-art meta-learning baselines.
>
> > **Q.** For DDP, is it possible to conduct some experiments of DDP on top of MAML?
>
> **A.** We first want to kindly remind the reviewer that we included (MAML-like) few-shot learning experiments in Appendix D (line 209). While we didn’t study the exact problem you mention, we investigated another interesting question of “can we improve the few-shot learning performance by increasing the model size given SAMA’s improved scalability?”, to which our preliminary answer is “yes”. This question is quite different from a majority of previous works where the model size is fixed and authors attempt to improve few-shot accuracy by designing new algorithms.
>
> As for the applicability of DDP to MAML, we will separately discuss two different aspects of MAML: 1) application and 2) algorithm. More specifically, MAML solves the *few-shot learning* application of meta-learning with the *iterative differentiation* algorithm.
> - Application: We expect DDP will have a limited impact on few-shot learning applications. DDP improves compute/memory efficiency by distributing samples in a mini-batch across multiple GPUs. However, by the nature of few-shot learning, this application usually has a very small batch size. Thus, distributing such a small batch across multiple GPUs would likely lead to only a modest improvement in compute/memory efficiency. However, meta learning has a lot of other applications besides few-shot learning, such as data optimization and hyperparameter optimization, which don’t necessarily have a small batch size. We expect our DDP scheme will have a meaningful impact on these applications.
> - Algorithm: We believe iterative differentiation in MAML would show a reduced compatibility with DDP compared to SAMA. As stated in Sec 3.1, the first-order nature (no Hessian computation) is crucial in achieving the improved DDP compatibility, but the iterative differentiation algorithm usually involves Hessian computation.
>
> ### **Clarity**
> > **Q.** It's straightforward for section 3.3, but it is a little bit difficult to follow for sections 3.1 and 3.2.
>
> **A.** While we are unable to edit the manuscript during the review period this year, we will improve our writing and the overall clarity in the camera ready version of our paper.
>
> > **Q.** How to understand $u$ in section 3.1?
>
> **A.** As mentioned in Eq. (2) (line 96), $u$ is the update function of gradient-based optimization. Here, we provide concrete examples of $u$ for several popular optimizers.
> - SGD: $u_t = g_t = g(\theta_t;\lambda_t)\quad$
> - SGD-M: $u = \beta m_{t-1} + g_t\quad$
> - Adam: $u = \frac{\beta_1 m_{t-1} + (1 - \beta_1) g_t}{\sqrt{\beta_2 v_{t-1} + (1 - \beta_2 g_t^2)}}$
>
> > **Q.** Data reweighting method in 4.1: what is the relationship between $w$ and $\lambda_r$, $c$ and $\lambda_c$? Are w and c networks? Also, it is helpful to cite similar papers.
>
> **A.** $w/c$ are respectively reweighting and label-correction functions (i.e. neural networks) parameterized by $\lambda_r/\lambda_c$. We cited a few relevant data reweighting and label-correction works in line 219, and will also add papers you suggested.
>
> > **Q.** Typos
>
> **A.** Thanks for pointing them out. We will fix them in the revision.
>
> We hope our response resolved most of your concerns, and helped you evaluate our work more positively. If you have other comments, we are more than happy to address them in the remaining review period.

---

> > ### Comment · Reviewer_YbqA · 2023-08-21
> > **post rebuttal comment**
> >
> > Thank authors' efforts for the rebuttal. And the response addresses most of my concerns. Please update the final version based on the response, especially for some clarification. For batch size of few shot learning, although the size is small, how about the batch size of tasks? Can we distribute tasks in parallel? Maybe this is also one potential solution? I raised my rating by 1.

---

> > > ### Author Response · Authors · 2023-08-22
> > > **Thanks for the response**
> > >
> > > We are glad that our rebuttal successfully addressed your concerns, and appreciate the score increase. Following the suggestion, we will incorporate this additional information in our final revision.
> > >
> > > **Task parallelism**
> > >
> > > Thanks for suggesting a very interesting idea. We believe it could potentially be achieved with some changes in implementation details (e.g. disabling DDP for base-level problems, each of which represents one task, as we don’t want to synchronize gradients across different tasks). Given that most MAML implementations handle each task sequentially, we don’t think such task parallelism will reduce the GPU memory usage, but it could still significantly improve the overall throughput (i.e. training speed). We will further investigate this in our future work.

---

### Official Review · Reviewer_fPdM · 2023-07-08

**Soundness:** 2 fair
**Presentation:** 3 good
**Contribution:** 2 fair
**Rating:** 5
**Confidence:** 4

**Summary:**

The authors explore the issues impacting the scalability of Gradient-based Meta-Learning (GBML), including high memory/compute costs, algorithmic instability, and poor support for distributed training. The causes identified are: the base Jacobian inversion, the absence of algorithmic adaptation for adaptive optimizers, and the requirement for a custom backward pass of meta gradient computation. Proposed solutions include: approximating the base Jacobian with an identity matrix, expanding the meta Jacobian through the chain rule, and creating a communication strategy leveraging the communication-computation overlap trick.

**Strengths:**

1. The application of meta-learning (bi-level optimization) is widespread in various aspects of deep learning, and considering the acceleration of meta-learning is an important direction.
2. Validating the effectiveness of meta-learning (bi-level optimization) on large-scale datasets and models is important, which can verify the ability of meta-learning to solve practical problems.
3. The manuscript is written quite clearly and is easy to read.


**Weaknesses:**

1. In the methodology section, the explanation of why an identity matrix can be used to approximate the base Jacobian is unclear and lacks necessary analysis. As for approximating the second-order derivative with the first-order derivative, the method is almost the same as DARTS. Distributed training naturally fits in a scenario where only the first-order derivative is used, and the authors did not provide any special design for it.
2. In the experiment section, the authors should focus on comparing the difference in training efficiency between the meta-learning acceleration framework proposed in this paper and other approximation methods. However, this is only reflected in Table 2, and there is no comparison with the approximation method proposed by DARTS, which is the method most closely related to this paper.
3. It would be beneficial for this work to provide some analysis demonstrating the distance of each approximate solution from the optimal solution, as this could provide more valuable guidance.


**Questions:**

1. Can you further explain why an identity matrix can be used to approximate the base Jacobian?
2. What is the innovative point of the distributed training proposed in this paper? What are the main benefits brought about by this innovation?
3. Can you provide some analysis demonstrating the distance of each approximate solution from the optimal solution?

Please see Weaknesses for details.

**Limitations:**

Please see Weaknesses.

---

> ### Author Rebuttal · Authors · 2023-08-09
>
> We thank you for the valuable feedback that will improve the quality of our work. We attempt to clarify and address your concerns regarding our work here and in the **global response**.
>
> ### **Comparison with DARTS**
> While we recognize the similarity, SAMA is different from DARTS in two major aspects.
> 1. Due to a lack of algorithmic adaptation in DARTS, SAMA and DARTS differ when an adaptive optimizer is used at the base level. As we pointed out in the paper, recent large models are by default trained with adaptive optimizers, and we empirically demonstrate in Table 1 that lacking this adaptation often leads to noticeable performance degradations. When SGD is used as a base optimizer, SAMA and DARTS become more similar, but they still have another important difference which we discuss in the next paragraph.
> 2. As stated in line 138-139, DARTS computes a meta Jacobian at initialization $\theta$, while SAMA computes it at (approximate) convergence $\theta^*$. Therefore, DARTS is closer to iterative differentiation while SAMA is implicit differentiation. This difference has two important implications in both memory/compute efficiency for scalable meta learning.
> - Memory: DARTS needs to keep the copies of both the initial parameter $\theta$ and the most recent parameter $\theta^*$ while SAMA only requires tracking $\theta^*$ (reference: the official implementation of DARTS, which indeed separately saves the copy of $\theta$). Therefore, DARTS incurs additional memory usage, worsening the memory bottleneck issue in scalable meta learning.
> - Compute: Compared to DARTS, SAMA more naturally allows for a larger unroll step, which essentially reduces the frequency of the meta-gradient computation, the most expensive operation in meta-learning. Given that the meta-objective is evaluated at the optimal base solution $\theta^*$, the quality of meta-gradient naturally depends on the quality of $\theta^*$ approximation. However, DARTS computes the meta-gradient at $\theta$, and therefore the approximation error of $\theta^*$ increases as we use larger unroll steps. Indeed, the original DARTS paper only used an unroll step of 1, while we were able to use an unroll step of 10 for our Transformer experiments (Sec 4.1 & 4.2). This significantly improves the overall training efficiency (i.e. throughput) of meta learning.
> To more clearly understand these benefits, we provide a quantitative analysis of memory/computation efficiency between SAMA and DARTS in an additional ablation study. The result can be found in Table 1 & 2 in the global response.
>
> ### **Justification of identity approximation**
> As shown in [17], approximating base Jacobian as identity can be understood as preconditioning meta-gradient. In other words, given the optimization path from $\theta$ to $\theta^*$, the identity base Jacobian trick in SAMA essentially approximates this optimization path with the *reverse* one step update (i.e. gradient ascent) from $\theta^*$. Similarly, in DARTS, which also adopts the identity base Jacobian trick, it approximates the optimization path with the one step update (i.e. gradient descent) from $\theta$.
> Following your suggestion, we additionally analyzed the effect of this approximate solution (i.e. the distance of the approximate solution obtained by SAMA from the optimal solution) on the meta-gradient computation and the final optimal meta solution ($\lambda^*$) in the “biased regression” setting, where the closed-form solution can be obtained analytically. In this experiment, we empirically show that the identity approximation still allows for the accurate estimation of meta-gradients and the optimal meta solution $\lambda^*$, even when the true base Jacobian is not an identity matrix. More detailed discussion is provided in the global response.
>
> [17] Fung et al., Jfb: Jacobian-free backpropagation for implicit networks. AAAI, 2022.
>
> ### **Benefits of SAMA’s distributed training scheme**
> To the best of our knowledge, our work is first to raise the problem of the communication efficiency in distributed meta learning, and propose an initial solution. Given recent advances in hardware, the communication overhead can quickly become a bottleneck for training efficiency in large-scale learning [31]. As stated in Sec 3.3, one meta gradient computation in SAMA involves 3 backward computations. If implemented naively, gradient synchronization can happen after each backward computation, whereas SAMA performs synchronization only once after the last backward computation. This can roughly reduce the communication cost by three times. In addition, most meta learning implementations including DARTS heavily rely on `torch.autograd.grad` instead of `torch.autograd.backward`. Unfortunately, autograd.grad doesn’t support communication optimization such as a communication-computation overlap, which is essential in large-scale learning as shown in [31]. In summary, in addition to simply applying distributed training to our first-order SAMA algorithm, we perform additional communication optimizations that (1) reduce communication cost by 3 times through sparse gradient synchronization, and (2) hide the remaining communication cost behind the computation through the mixed use of autograd.grad and autograd.backward. In doing so, we were able to maximize training efficiency of distributed meta-learning.
>
> [31] Li et al., Pytorch distributed: Experiences on accelerating data parallel training. VLDB, 2021.
>
> We hope our response resolved most of your concerns, and helped you evaluate our work more positively. If you have other comments, we are more than happy to address them in the reviewer-author discussion period.

---

> > ### Comment · Reviewer_fPdM · 2023-08-13
> >
> > Thank you for your response. This reply provided additional information, and I am willing to raise my score.

---

> > > ### Author Response · Authors · 2023-08-14
> > > **Thanks for the response**
> > >
> > > We are glad that our rebuttal alleviated your concerns. We will incorporate these additional information into our final revision. Thanks again for your time and effort!

---

### Official Review · Reviewer_BWd8 · 2023-07-09

**Soundness:** 2 fair
**Presentation:** 2 fair
**Contribution:** 2 fair
**Rating:** 5
**Confidence:** 3

**Summary:**

This paper tries to scale current meta learning algorithm and make scalable meta learning practical. Specially, the authors propose a novel algorithm SAMA from the perspective of algorithm and system, which can support arbitrary optimizers in the base level of meta learning and reduce the computation cost. The experimental results illustrate that the proposed method SAMA can improve the throughput and meanwhile reduce the computation cost in large-scale benchmarks. For example, they try to evaluate the performance of SAMA on text classification with large language models, such as  BERT and RoBERTa.

**Strengths:**

This paper focus on great direction and how to scale meta learning is very important, especially when large language models are so popular now.
The proposed method is also easy to follow, especially the authors try to solve the problem from the perspectives of algorithm and system.

**Weaknesses:**

This paper mainly focus on scaling and maybe you should provide more experimemtal analysis about scaling, such as from small mode to large model, from 1 gpu to more gpus. Maybe you can provide more results of your proposed method and baseline about scaling.
The most experiments concentrate on nlp tasks. However, we know there are also dome large models on other tasks, such as computer vision. Therefore, maybe you should provide more results on other tasks to illustrate the generality of SAMA.
I think the contribution from system is a little weak and the workflow can be directly implemented with original PyTorch.

**Questions:**

In section 3.1 and 3.2, you provide two solutions about two important problems. I would like to ask do you try to compare your proposed method with the related works that also try to solve these problems?

---

> ### Author Rebuttal · Authors · 2023-08-09
>
> Thank you for taking the time to review our work, and for the useful feedback. We address the comments and questions raised in your review below and in the **global response**.
>
> ### **Additional scalability analysis**
> > **Q.** This paper mainly focus on scaling and maybe you should provide more experimental analysis about scaling, such as from small mode to large model, from 1 gpu to more gpus. Maybe you can provide more results of your proposed method and baseline about scaling.
>
> **A.** We agree with the value of these experiments, and want to highlight the three scalability analyses in our paper — Table 2 & Figure 1 (bottom left): memory/throughput analysis on the noisy finetuning task, Figure 1 (bottom right): memory vs model size analysis on the continued pretraining task, and Figure 4 (Appendix D): scale-accuracy analysis on the image few-shot learning task. Given your suggestion, we additionally performed a more extensive ablation study on two datasets from the Wrench benchmark, and presented the results in the global response. From the results, you can clearly see the effects/benefits of each component in SAMA for scalable meta-learning.
>
> ### **Other domains than NLP**
> > **Q.** The most experiments concentrate on nlp tasks. However, we know there are also some large models on other tasks, such as computer vision. Therefore, maybe you should provide more results on other tasks to illustrate the generality of SAMA.
>
> **A.** We want to kindly remind the reviewer that we included two computer vision experiments, namely 1) data pruning on ImageNet/CIFAR datasets and 2) (MAML-like) few-shot image classification, respectively in Sec. 4.3 and Appendix D of our paper. With these two experiments, we tried our best to demonstrate the generality of SAMA across different domains. We admit that there are a lot of other interesting applications and domains to explore, though in this paper we aimed to prioritize the best subset of experiments given the 9-page space limit.
>
> ### **Distributed training**
> > **Q.** I think the contribution from system is a little weak and the workflow can be directly implemented with original PyTorch.
>
> **A.** To the best of our knowledge, our work is the first to raise the problem of efficient distributed meta learning as well as provides an initial solution to it. While our distributed training solution may look simple in retrospect, we noticed that most existing works on meta learning are limited to a single-GPU setup. For the remaining few works that combine distributed training and meta learning, we found that existing implementations are either incorrect (e.g. no proper gradient synchronization) or do not provide communication optimization.
> In contrast, as stated in Sec 3.3, our DDP strategy (1) reduces communication cost by 3 times through sparse gradient synchronization, and (2) hides the remaining communication cost behind the computation through our simple yet smart implementation trick.
>
> If you have any other comments that could make our paper stronger, we are more than happy to discuss them in the remaining review period. Thanks again for your reviewing effort!

---

### Official Review · Reviewer_2sgt · 2023-07-10

**Soundness:** 3 good
**Presentation:** 3 good
**Contribution:** 2 fair
**Rating:** 5
**Confidence:** 3

**Summary:**

The paper addresses the challenges of scalability in meta learning by introducing SAMA, a novel approach that combines advancements in implicit differentiation algorithms and systems. SAMA demonstrates improvements in computational efficiency and memory consumption compared to other baseline algorithms, and it showcases practical applicability in language and vision domains through experiments with large language models and image classification tasks.

**Strengths:**

1. Originality: The paper introduces a novel approach to address the scalability challenges of meta learning. By combining advancements in implicit differentiation algorithms and systems, it proposes a solution that supports arbitrary optimizers in meta learning while reducing computational burden.
2. Quality: The paper demonstrates a level of quality in terms of the overall idea and experimental evaluation. By evaluating their method on a few benchmarks, including language models and image classification tasks, the authors provide a good assessment of its performance.
3. Clarity: The key contributions, such as the introduction of SAMA and its evaluation of various benchmarks, are presented in a concise manner. The background and related work sections also help in properly positioning the paper in the literature.
4. Significance: The paper addresses the long-standing challenge of scalability in meta learning. It is able to tackle a variety of high-dimensional inductive biases of large-scale learning - for instance in optimizing large language models as a very relevant open problem to the community.

**Weaknesses:**

In general, the main weaknesses that I noticed are in the design of the experimental section:
1. Comparison with State-of-the-Art: The paper could benefit from a more comprehensive comparison with state-of-the-art meta learning approaches. Is it possible to compare it to well-established meta-learning algorithms, such as MAML?
2. Lack of ablations: More importantly, the paper seems to lack an ablation study. Including such comparisons would provide a more thorough understanding of the model's effectiveness and highlight more clearly its advantages over existing methods. It would be appreciated if the authors identify separate components of their method and ablate them w.r.t the full method.

**Questions:**

I would appreciate it if the authors address the following questions:
1. How does the inconsistency between the assumed vanilla SGD optimizer and the actual optimizer used in large models affect the computation of the meta gradient? Could you provide more insights into the inconsistencies and their consequences in terms of training instabilities and reduced performance in meta learning?
2. The paper mentions that solutions proposed in data-centric AI works to improve training data quality often rely on hand-designed heuristics. Could you elaborate on the specific limitations or drawbacks of existing approaches based on hand-designed heuristics? How does the proposed meta learning approach address or overcome these limitations?
3. In section 4.1, are there any specific techniques or algorithms employed within the meta learning framework to optimize the noisy training data? How does the proposed approach leverage meta learning to adaptively update and improve the quality of the labels generated by weak labeling functions?
4. What are the key differences between vanilla SGD and adaptive optimizers like Adam in terms of their impact on the fixed point condition (mentioned in section 3.2 line 149)?

**Limitations:**

The authors have provided a sufficient discussion of the limitations of their work in section 6. It would be beneficial for the authors to further elaborate on these aspects, beyond language models, to ensure a comprehensive analysis of any potential societal impacts.

---

> ### Author Rebuttal · Authors · 2023-08-09
>
> We appreciate your positive review and valuable comments. We strive to address concerns and questions that you raised below and in the **global response**.
>
> ### **Ablation Study & SOTA comparison**
> Though we didn’t explicitly frame it as an “ablation study”, the effectiveness of each component in SAMA can be understood from Table 1 & 2 in our paper. Below, we directly discuss each component of SAMA based on Table 1 & 2. We also provide a unified ablation study result table in the global response.
>
> **Base Jacobian inverse**
>
> *tl;dr* Identity approximation of base Jacobian significantly improves memory/compute efficiency (Table 2).
>
> Our baseline algorithms in Table 2 (and Figure 1), Neumann and CG, are both state-of-the-art implicit differentiation meta learning algorithms that attempt to approximate base Jacobian inverse as accurately as possible with multiple Hessian-vector product operations, instead of approximating it with an identity matrix as in SAMA. As shown in Table 2, due to an expensive second-order gradient computation involved in Hessian-vector products, these methods demonstrate much poorer throughput and GPU memory usage than SAMA, both of which are the major bottlenecks in scalable meta learning. In our global response, we additionally demonstrate that this identity approximation also has a minimal impact on accuracy by comparing SAMA against Neumann and CG in terms of accuracy.
>
> **Algorithmic adaptation for adaptive optimizer**
>
> *tl;dr* Algorithmic adaptation significantly improves accuracy (Table 1) at the minimal memory/compute cost (Table 2).
>
> The main goal of this work is to devise a *(1) memory/compute efficient* meta learning algorithm that *(2) achieves good performance/accuracy*. In Table 1, SAMA consistently achieves better accuracy than SAMA-NA, which lacks algorithmic adaptation. Moreover, Table 2 shows that SAMA achieves a comparable memory/compute efficiency as SAMA-NA.
>
> **Distributed training**
>
> *tl;dr* Both GPU memory usage and throughput improve consistently as computations are distributed across more GPUs (Table 2).
>
> We agree that it may not be straightforward for readers to clearly understand the effectiveness of each component when the results are spread across two separate tables. Hence, in the global response, we provide a unified table for Wrench experiments, including a comparison with state-of-the-art meta-learning baselines.
>
> ### **Questions**
> > Q. How does the inconsistency between the assumed vanilla SGD optimizer and the actual optimizer used in large models affect the computation of the meta gradient? Could you provide more insights into the inconsistencies?
>
> Roughly speaking, once base Jacobian is approximated as identity, the meta gradient formulation follows $g_{meta}=-\frac{\partial u}{\partial \lambda}\cdot\frac{\partial L_{meta}}{\partial \theta^*}=-\frac{\partial}{\partial \lambda}(u\cdot\frac{\partial L_{meta}}{\partial \theta^*})$. Thus, meta gradient descent essentially maximizes the inner product between the base update vector $u$ and meta gradient w.r.t base parameters $\theta$. This way, performing base updates with $u$ not only decreases the base loss, but also *maximally* decreases the meta loss. We note that the update direction of the base problem $u$ is dependent on its optimizer. Thus, we believe that reflecting this base optimizer choice would lead to improved meta learning performance in the end by better aligning $u$ and $\frac{\partial L_{meta}}{\partial \theta^*}$.
>
> > Q. The paper mentions that solutions proposed in data-centric AI works to improve training data quality often rely on hand-designed heuristics.
> Could you elaborate on the specific limitations of existing approaches based on hand-designed heuristics?
>
> The benefits of meta-learning approaches to data-centric AI can be most clearly seen in our dataset pruning experiments (Sec 4.3). Specifically, heuristics-based methods (e.g. EL2N, forgetting) rank each training sample using some heuristics, such as forgetting counts and gradient norm at initialization, *in the hope* that these heuristics would indeed capture the importance weight of each sample. On the other hand, our meta-learning approach directly optimizes the importance weight of each sample in a way that the resulting model trained with these learned importance weights minimizes the original training loss. Indeed, our experiment results in Fig. 3 clearly shows that our meta-learning approach outperforms all heuristics-based methods on both small-/large-scale datasets.
>
> > Q. In section 4.1, are there any specific techniques or algorithms employed within the meta learning framework to optimize noisy training data?
>
> There are abundant meta-learning works for tackling various data issues (e.g. noisy labels, class imbalance). A few examples are Meta-Weight-Net [50], Learning-to-Reweight [48], and Meta-Label-Correction [60]. We adopt some of the architecture designs from these works, but replace the meta-gradient computation algorithm with SAMA, instead of iterative differentiation or truncated backpropagation, to improve the scalability.
>
> > Q. What are the key differences between vanilla SGD and adaptive optimizers like Adam in terms of their impact on the fixed point condition?
>
> Most gradient-based optimizers, including SGD and Adam, share the same fixed point condition of $\frac{\partial L_{base}}{\partial \theta^*} = 0$. However, most gradient-based meta learning in practice approximates $\theta^*$ with only a few gradient updates, and thus this “zero gradient” condition is unlikely met. Therefore, we hypothesize that we in reality need to pay more attention to the alignment between $u$ and $\frac{\partial L_{meta}}{\partial \theta}$ discussed above, where $u$ is dependent on the base optimizer.
>
> We hope our response resolved most of your concerns, and helped you evaluate our work more positively. If you have other comments, we are happy to address them in the reviewer-author discussion period.

---

> > ### Comment · Reviewer_2sgt · 2023-08-21
> >
> > Thanks for the response. The concerns are mostly addressed, and I will keep my initial score.

---

> > > ### Author Response · Authors · 2023-08-21
> > >
> > > We are glad that most of your concerns are addressed. Thanks for your reviewing effort again.

---

### Author Rebuttal · Authors · 2023-08-09

We first want to express our gratitude to all reviewers for their reviewing efforts. In our global response, we address two issues raised by reviewers: 1) Ablation study and 2) (empirical) justification of the identity base Jacobian approximation.

### **Ablation Study & SOTA comparison**
While the effectiveness of each component in SAMA (i.e. base Jacobian inverse, algorithmic adaptation for the adaptive optimizer, and efficient distributed training) can be collectively understood from Table 1 & 2, several reviewers asked for a more direct ablation study. Therefore, we provide below a unified table for the ablation study. In detail, our experiment settings are:

- Datasets: 1) AGNews and 2) IMDB from the Wrench benchmark (Sec 4.1)

- Baselines: 1) fine-tuning (no meta-learning), 2) iterative differentiation (e.g. MAML), 3) conjugate gradient (e.g. iMAML), 4) Neumann series, 5) DARTS, 6) SAMA-NA (no algorithmic adaptation)

**Table 1. Ablation results on AGNews**

|                  | Base Jacobian | Algo Adapt | Distributed | Accuracy | Throughput | Memory |
|:----------:|:---------:|:-------:|:-------:|:------:|:-------:|:-----:|
| Finetuning (Baseline)  | x | x | x | 85.79 | 169.16 | 7.77 |
| Iterative Differentiation (MAML) | x | x | x | 85.78 | 28.07 | 22.94 |
| Conjugate gradient (iMAML) | x | x | x | 86.78 | 65.14 | 22.03 |
| Neumann series | x | x | x | 86.65 | 67.03 | 19.70 |
| DARTS | o | x | x | 86.36 | 43.69 | 10.81 |
| SAMA-NA | o | x | x | 86.55 | 137.90 | 10.30 |
| SAMA | o | o | x	| **89.05** | 134.56 | 11.12 |
| SAMA (2 GPUs) | o | o | o | **88.85** | 226.27 | 8.00 |
| SAMA (4 GPUs) | o | o | o | **89.02** | **298.28** | **6.46** |

**Table 2. Ablation results on IMDB**

|                  | Base Jacobian | Algo Adapt | Distributed | Accuracy | Throughput | Memory |
|:--------:|:---------:|:------:|:--------:|:------:|:-------:|:-----:|
| Finetuning (Baseline)  | x | x | x | 78.16 | 144.39 | 6.60 |
| Iterative Differentiation (MAML) | x | x | x | 80.25 | 24.24 | 22.03 |
| Conjugate gradient (iMAML) | x | x | x | 81.01 | 56.27 | 21.92 |
| Neumann series | x | x | x | 79.92 | 57.85 | 19.75 |
| DARTS | o | x | x | 80.47 | 37.53 | 10.35 |
| SAMA-NA | o | x | x | 81.92 | 117.86 | 9.93 |
| SAMA | o | o | x	| **84.31** | 116.94 | 10.84 |
| SAMA (2 GPUs) | o | o | o | **85.18** | 196.48 | 7.84 |
| SAMA (4 GPUs) | o | o | o | **84.19** | **263.74** | **6.39** |

From our extended ablation study, it can be seen that 1) an identity approximation of base Jacobian significantly improves memory/compute efficiency, 2) algorithmic adaptation improves meta learning performance at the minimal compute/memory cost, and 3) our communication-optimized distributed training further improves compute/memory efficiency.

### **Analysis on Identity Approximation of the Base Jacobian**
As obtaining the closed form solution of the Hessian is impossible in almost all deep learning problems, we study the soundness of the identity approximation of base Jacobian in the simpler “biased regression” setting [1], for which the bilevel optimization formulation is as follows:
$$
\lambda^* = \arg\min_{\lambda} \Vert X’w^*(\lambda) - y’\Vert^2
$$

$$
w^*(\lambda) = \arg\min_w \Vert Xw - y\Vert^2 + \beta\Vert w - \lambda \Vert^2
$$
Given the above formulation, the closed-form solutions for the base Jacobian, the meta-gradient $g_{\lambda}$, and the optimal meta solution $\lambda^*$ are:

- base Jacobian $= X^TX + \beta I$
- $g_{\lambda} = \beta (X^TX + \beta I)^{-1}(X’^TX’w^* - X’^Ty’)$, where $w^* = (X^TX + \beta I)^{-1}(X^Ty + \beta\lambda)$
- $\lambda^* = (A^TA)^{-1}A^Tb$, where $A=\beta X’(X^TX + \beta I)^{-1}), b=y’ - X’(X^TX + \beta I)^{-1}X^Ty$

We set $\beta=0.1$ and perform 100 meta updates, and measure 1) the cosine similarity between the ground truth $g_{\lambda}$ and the meta gradient $g_{approx}$ obtained with our approximation, and 2) the L2 distance between the current meta parameter $\lambda_t$ and the optimal solution $\lambda^*$ at each time step $t$. For a more thorough analysis, we also compute these two metrics for other meta gradient algorithms that explicitly approximate base Jacobian inverse with conjugate gradient and Neumann series. In the table below, we provide the metric obtained from several different time steps. *The visual plots for all time steps are provided in the attached pdf file at the bottom of this response.*

**Table 3.** $cos(g_{\lambda}, g_{approx})$ **result**

|     	| t=0	| t=10   | t=20   | t=50   | t=100  |
|---------|--------|--------|--------|--------|--------|
| CG  	| 0.9995 | 0.9994 | 0.9989 | 0.9997 | 0.9319 |
| Neumann | 0.9957 | 0.9952 | 0.9949 | 0.9948 | 0.9363 |
| SAMA	| 0.9843 | 0.9818 | 0.9815 | 0.9861 | 0.9321 |

**Table 4.** $\Vert \lambda^* - \lambda_t\Vert_2$ **result**

|     	| t=0	| t=10   | t=20   | t=50   | t=100  |
|---------|--------|--------|--------|--------|--------|
| CG  	| 3.6752 | 1.8959 | 0.9507 | 0.1953 | 0.0160 |
| Neumann | 3.6972 | 2.6116 | 1.3468 | 0.2823 | 0.0216 |
| SAMA	| 3.6856 | 2.1537 | 0.6952 | 0.1966 | 0.0184 |

From Table 3 & 4 (and plots in the attached PDF), we can clearly see that 1) while slightly less accurate than second-order algorithms like CG, SAMA still achieves a high directional alignment with the ground truth meta-gradient, and 2) SAMA also achieves a stable convergence to the optimal solution at a comparable speed. We hope this result corroborates the soundness of our identity approximation for the base Jacobian.

[1] Grazzi et al., On the Iteration Complexity of Hypergradient Computation. ICML, 2020.

Note: We used smaller $\beta$ than in the original paper (1 vs 0.1), to amplify the “non-identitiness” of the base Jacobian.

We will include all of the above results in the camera ready version of our paper. If reviewers have any further questions regarding these additional experiments, we are happy to discuss them during the author-reviewer discussion period. Thanks again!

---

### Decision · Program_Chairs · 2023-09-21

**Decision:**

Accept (poster)

**Comment:**

The topic of practical meta learning is appreciated, the approach using implicit differentiation is novel and of interest and the demonstrated computational benefits are convincing. The author provided responses to several reviewer comments have been well received, and after author/reviewer discussion all five reviewers are supportive of acceptance. The AC agrees and encourages the authors to include all the promised additions and experiments in the camera-ready paper.